# Robotic-Assisted versus Laparoscopic Left Hemicolectomy—Postoperative Inflammation Status, Short-Term Outcome and Cost Effectiveness

**DOI:** 10.3390/ijerph191710606

**Published:** 2022-08-25

**Authors:** Anna Widder, Matthias Kelm, Joachim Reibetanz, Armin Wiegering, Niels Matthes, Christoph-Thomas Germer, Florian Seyfried, Sven Flemming

**Affiliations:** 1Department of General, Visceral, Transplantat, Vascular and Pediatric Surgery, Center of Operative Medicine (ZOM), University Hospital of Wuerzburg, 97080 Wuerzburg, Germany; 2Department of Biochemistry and Molecular Biology, University of Wuerzburg, 97070 Wuerzburg, Germany; 3Comprehensive Cancer Center Mainfranken, University of Wuerzburg Medical Centre, 97080 Wuerzburg, Germany

**Keywords:** robotic surgery, colon resection, postoperative inflammation, cost-effectiveness, left hemicolectomy

## Abstract

Robotic-assisted colon surgery may contain advantages over the laparoscopic approach, but clear evidence is sparse. This study aimed to analyze postoperative inflammation status, short-term outcome and cost-effectiveness of robotic-assisted versus laparoscopic left hemicolectomy. All consecutive patients who received minimal-invasive left hemicolectomy at the Department of Surgery I at the University Hospital of Wuerzburg in 2021 were prospectively included. Importantly, no patient selection for either procedure was carried out. The robotic-assisted versus laparoscopic approaches were compared head to head for postoperative short-term outcomes as well as cost-effectiveness. A total of 61 patients were included, with 26 patients having received a robotic-assisted approach. Baseline characteristics did not differ among the groups. Patients receiving a robotic-assisted approach had a significantly decreased length of hospital stay as well as lower rates of complications in comparison to patients who received laparoscopic surgery (n = 35). In addition, C-reactive protein as a marker of systemic stress response was significantly reduced postoperatively in patients who were operated on in a robotic-assisted manner. Consequently, robotic-assisted surgery could be performed in a cost-effective manner. Thus, robotic-assisted left hemicolectomy represents a safe and cost-effective procedure and might improve patient outcomes in comparison to laparoscopic surgery.

## 1. Introduction

Robotic surgery has gained momentum worldwide with fast-growing expansions in various disciplines during the last decade [1]. While a minimal-invasive approach compared to open surgery has been shown to be vastly superior regarding short-term outcomes, increasing evidence also demonstrates comparable long-term results for oncological and non-oncological indications [2,3]. Despite robotic surgery offering additional benefits, including elimination of natural tremor with significantly improved visualization, better ergonomics and enhanced dexterity, robust evidence demonstrating that robotic surgery can further increase the advantages of laparoscopic surgery is lacking.

In the field of visceral surgery, robotic-assisted surgery may contain some benefits in complex oncological procedures, such as esophageal and rectal surgery [4]. For the latter, an advantage of the robotic compared to the laparoscopic approach was demonstrated in the ROLARR-Trial regarding rates of conversions. However, the quality of resection was comparable, while long-term oncological results are still lacking [5]. Nevertheless, randomized trials comparing laparoscopic vs. robotic-assisted colorectal surgery are missing, which is especially important due to the high morbidity and mortality following colorectal resections [6,7]. Despite the lack of evidence for the superiority of robotic-assisted surgery, numbers of robotic procedures are rapidly increasing, and novel fields, including colon, gastric and pancreatic surgery, are evolving [1]. While there are consistent improvements in the technical and surgical aspects of robotic-assisted surgery, criticisms of those mostly non-evidence-based developments remain relevant and are not deniable. While operating times are significantly prolonged for robotic-assisted approaches, enhanced costs of robotic-assisted surgery in comparison to laparoscopic surgery without clear evidence for an improved short- and long-term outcome remain another major issue. Therefore, the advantage of robotic surgery in comparison to laparoscopic surgery for standardized minimally invasive procedures in visceral surgery still needs to be determined [8].

To assess and evaluate the current role and future potential of robotic-assisted surgery in an established minimal-invasive operation, we analyzed postoperative inflammation, short-term outcomes and costs for all consecutive patients who received left-sided hemicolectomy due to benign and malign diseases before and after re-introduction of robotic-assisted resection at our hospital.

## 2. Materials and Methods

### 2.1. Study Population

All consecutive patients who received minimal-invasive left hemicolectomy due to chronic relapsing diverticulitis, left-sided colon cancer UICC I–III and endometriosis at the Department of Visceral Surgery of the University Hospital of Wuerzburg in 2021 were prospectively included in this study. All surgeries were performed by a senior surgeon (n = 3 surgeons) in line with international standards, and reconstruction was done as end-to-end stapler-anastomosis. Two of these three colorectal surgeons performing the laparoscopic resections also did the robotic operations. Robotic colorectal surgery at the Department of Visceral Surgery was re-launched in May 2021. The robotic surgical experience of the 2 robotic surgeons was limited to being 1st assistant.

All patients were divided into two groups depending on the type of surgery (laparoscopic versus robotic-assisted). While in the beginning, all patients were operated laparoscopically, robotic-assisted left hemicolectomy was introduced in May 2021, with most of the patients receiving robotic-assisted surgery for the rest of the observational period. Patients were not specifically selected for laparoscopic or robotic-assisted surgery. Sociodemographic and clinicopathological data, including diagnosis, history of disease and co-morbidities, were collected for each patient from patient records. In addition, surgical data, including the rate of conversion, operating time as well as complications, were also analyzed. Furthermore, postoperative inflammation was evaluated by leukocyte count and C-reactive protein levels.

### 2.2. Outcome

The primary endpoint was defined as the length of hospital stay. Secondary endpoints were postoperative complications within 30 days, including MTL30 [9]. Furthermore, serum levels of leukocytes and C-reactive protein (CRP) were collected on postoperative days 1, 3 and 5.

### 2.3. Cost Effectiveness

To analyze the cost effectiveness between laparoscopic and robotic-assisted left hemicolectomy in our cohort, calculations were performed as published previously [10]. Briefly, costs were analyzed per patient and included procedure-related costs as well as costs during the hospital stay. For costs of operating time and hospital stay, average values were used [11]. Importantly, costs for materials and employees, as well as maintenance and acquisition of the robotic system, were also included in the analysis according to internal and previously published data [12].

### 2.4. Statistical Analysis

Statistical analysis was performed using IBM SPSS Statistics for Windows, version 28.0 (IBM Corp., Armonk, NY, USA). Descriptive data are presented as median with range or total numbers with percentage. Differences in patient characteristics were assessed by the Chi-squared test, Fisher’s exact test, or ANOVA test according to data scale and distribution. Multivariate analyses of variance were performed by using MANOVA. Statistical relevance was considered for a *p*-value < 0.05.

### 2.5. Ethical Approval

Ethical approval for this study was obtained from the Ethics Committee of the University of Wuerzburg, Germany.

## 3. Results

### 3.1. Patient Cohort

In this single-center study, 61 patients received minimal-invasive left hemicolectomy in 2021 at the Department of Surgery at the University Hospital of Wuerzburg. Of those, 35 patients underwent laparoscopic surgery, whereas 26 patients were operated on in a robotic-assisted manner. As presented in Table 1, both groups did not differ regarding age, BMI, ASA classification and Charlson Comorbidity Index (CCI). However, more male patients were operated on laparoscopically, whereas robotic-assisted surgery was predominantly performed on women. Further analysis revealed no significant differences for co-morbidities between both patient groups. Similarly, no differences were observed between the laparoscopic and robotic-assisted cohorts regarding the indication for surgery.

### 3.2. Postoperative Outcome

Operating times in patients receiving robotic-assisted left hemicolectomy were significantly longer compared to patients who were operated on laparoscopically (254 min versus 173 min; *p* = 0.001). However, rates of conversion were comparable between both groups (11.4% versus 11.5%; *p* = 0.989). Length of hospital stay, the primary endpoint of patients receiving robotic-assisted surgery, was significantly reduced compared to patients who underwent laparoscopic resection (6 versus 10 days, *p* = 0.025) (Table 1). In a multivariate analysis, this trend was confirmed without reaching statistical significance (Table 2). Similarly, laparoscopic surgery tended to increase levels of postoperative complications such as anastomotic leakage (8.6% versus 0%, *p* = 0.126) (Table 3).

### 3.3. Development of Serum Levels of Leukocytes and C-Reactive Protein (CRP) Postoperatively

Serum levels of leukocytes and C-reactive protein (CRP) were analyzed during the postoperative course to evaluate differences in inflammatory responses of patients receiving minimally invasive surgery. While on postoperative day 3, serum levels of CRP trended to be lower following robotic-assisted surgery in comparison to laparoscopic resection, CRP levels were significantly decreased on postoperative day 5 in patients who received robotic-assisted colon resection (Table 4; Figure 1). No differences in serum levels of leukocytes were observed between both groups. In a multivariate analysis, decreased levels of CRP at postoperative days 3 and 5 were identified as independent prognostic factors (Table 2).

### 3.4. Cost Effectiveness

Costs for both groups were calculated, and differences are presented in Table 5 and Figure 2. Interestingly, despite higher costs for the robotic-assisted procedure itself (EUR 1412.83 versus EUR 2017.17), overall costs were lower when patients were operated on in a robotic-assisted manner (EUR 6796.37 versus EUR 6559.89), with a cost-effectiveness of EUR 236.48 per patient (Figure 2). The main difference regarding cost-effectiveness between both groups was seen in the costs of the hospital stay, which were much higher for laparoscopic surgery (EUR 3455.50 versus EUR 2073.20) (Table 5). Importantly, costs for the robotic system, including acquisition and maintenance, were calculated per patient over a period of ten years with eight operating days per month.

## 4. Discussion

Robotic-assisted surgery represents a rapidly evolving field in visceral surgery. While the advantages of minimal-invasive procedures over open surgery are well established, robust evidence about the benefits of the robotic-assisted approach in comparison to laparoscopic surgery is still underrepresented [8]. In addition, robotic-assisted surgery has been shown to contain higher costs which are widely seen as the major disadvantage, thus detaining further implementation of this technique. Therefore, we performed an analysis of our non-selective cohort comparing postoperative outcomes of patients receiving either laparoscopic or robotic-assisted resection to address those questions. Based on our data, we demonstrate that patients have a significantly decreased length of hospital stay as well as a trend for decreased rates of complications, including a lower systemic inflammatory response following robotic-assisted left hemicolectomy, which results in improved cost-effectiveness for laparoscopic and robotic-assisted surgery.

After the re-implementation of robotic-assisted left hemicolectomy at our department, robotic-assisted surgery led to a significantly decreased length of hospital stay (6 versus 10 days, *p* = 0.001) (Table 1). Similarly, rates of complications, including anastomotic leakage, trended to be lower for robotic-assisted surgery in comparison to laparoscopic surgery without reaching statistical significance due to low overall numbers (Table 3). Thus, to draw a final conclusion as to whether the robotic approach leads to lower morbidity, further studies are needed. Despite the increased operating time for robotic-assisted procedures based on the mandatory learning curve for robotic surgery, conversion rates demonstrated no differences between both groups (11.4% versus 11.5%, *p* = 0.989). Operative time and conversion rates in our study were comparable to another comparable study including only left hemicolectomies [13]. Importantly, no preoperative patient selection was performed, which is demonstrated by comparable patient characteristics between both groups, including BMI and co-morbidities (Table 1). In line with that, almost all patients were operated on in a robotic-assisted manner after the implementation of the technique in our department.

In general, the advantages of robotic surgery have been extensively described elsewhere, including improved visualization and filtration of physiological tremors. While a Danish cohort study analyzed both approaches for right colectomy and found similar morbidity between both techniques [14], Tschann et al. demonstrated, in a recent meta-analysis, the benefits of robotic-assisted right hemicolectomy on conversion rates as well as hospital stay with similar oncological long-term outcomes [15]. Furthermore, another meta-analysis by Cuk et al. confirmed the potential advantages of robotic-assisted colon surgery on surgical morbidity and efficacy [16]. However, only a few studies have analyzed the effects of robotic-assisted left hemicolectomy on patient outcomes in comparison to laparoscopic surgery. In line with the results of the previous studies but with the focus on left hemicolectomy only, Giordano et al. showed decreased rates of complications and readmissions for robotic-assisted surgery in comparison to laparoscopic surgery in sigmoid resection [17]. Those results were confirmed by a systematic review which demonstrated that rates of overall complications, anastomotic leakage and wound infections were decreased following robotic-assisted left hemicolectomy [18].

To further assess potential mechanisms of the beneficial effect of robotic-assisted surgery on short-term patient recovery, we analyzed serum levels of inflammation such as leukocytes and CRP to evaluate the systemic inflammatory response. Indeed, levels of CRP were significantly lower during the postoperative course for patients who were operated on in a robotic-assisted manner in comparison to patients who received laparoscopic resection (Table 4). Importantly, decreased levels of CRP on postoperative days 3 and 5 were identified as independent prognostic factors for a reduced length of hospital stay in a multivariate analysis (Table 2). While there was no direct statistically significant effect of the surgical technique on the length of hospital stay in a multivariate analysis, robotic-assisted surgery significantly reduced CRP levels on days 3 and 5 (Table 4). Thus, our results demonstrate that the decreased length of hospital stay was mainly because of a reduction in systemic inflammation, which was clearly associated with robotic-assisted procedures. Decreased rates of systemic inflammation in robotic-assisted surgery could be explained by filtration of physiological tremors and more stable instruments resulting in less stress on the abdominal wall, including decreased peritoneal affection. Improved levels of inflammatory markers postoperatively might provide a hint to a potential explanation for the beneficial patient outcome. Our observation of a decreased inflammatory stress response is in line with a previous study demonstrating a similar postoperative course of CRP levels between laparoscopic and robotic surgeries [19]. However, to confirm those results, prospective studies such as the SIRIRALS-Trial are necessary before final conclusions on potential mechanisms can be made [20].

High costs remain to be one of the major criticisms about the implementation of robotic systems in surgery. Therefore, and in addition to previous studies, we also performed a cost analysis in our cohort. Interestingly, cost-effectiveness was comparable between both groups, but a relevant cost reduction of EUR 236.48/case was seen for robotic-assisted resections (Figure 2). This effect was mainly due to a reduced length of hospital stay and a trend for decreased rates of complications (EUR 3455.50 versus EUR 2073.20), thus compensating for enhanced costs of acquisition and maintenance as well as during operation due to the prolonged operating time (EUR 1412.83 versus EUR 2017.17) (Table 5). While this is an interesting observation, it should be taken into account that the projected costs represent average amounts, while costs per operating minute can vary significantly between hospitals and operating theatres. However, costs of acquisition, maintenance and disposables were included in our analysis and calculated for eight operating days per month only. Therefore, the presented data represent a major aspect of our study since more frequent usage of the robotic system can further decrease potential costs per patient. With the improvement in patient recovery seen by decreasing postoperative complications and the length of hospital stay, investing in a frequently used robotic system can be valuable and cost-effective for surgical departments based on our analysis in regards to colectomy.

The major limitations of our study are its retrospective character as well as the single-center design. In addition, the calculated costs for operating time and length of hospital stay are based on average costs and can vary from hospital to hospital depending on various parameters. However, our calculations represent a high validity and are based on previously published and established models. Furthermore, no patient selection was performed, with almost all patients being operated on in a robotic-assisted manner following the implementation of the technique in our department. Finally, the learning curve for the robotic approach must be considered; thus, the extended operating time and the conversion rate might decrease over time, resulting in a further cost reduction. Thus, the cases of conversion occurred at the beginning of the learning curve, supporting previous studies that show that the rate of conversion is even lower in more complex colorectal operations compared to the laparoscopic approach [21].

## 5. Conclusions

In conclusion, we demonstrated in a single-center study that robotic-assisted left hemicolectomy resulted in decreased length of hospital stay and trended to lower rates of complications, leading to improved cost-effectiveness. Based on that, our study supports the ongoing implementation of robotic-assisted colon surgery, but further prospective studies are necessary to confirm the results and to evaluate the mechanistic background of a potentially decreased inflammatory stress response.

## Figures and Tables

**Figure 1 ijerph-19-10606-f001:**
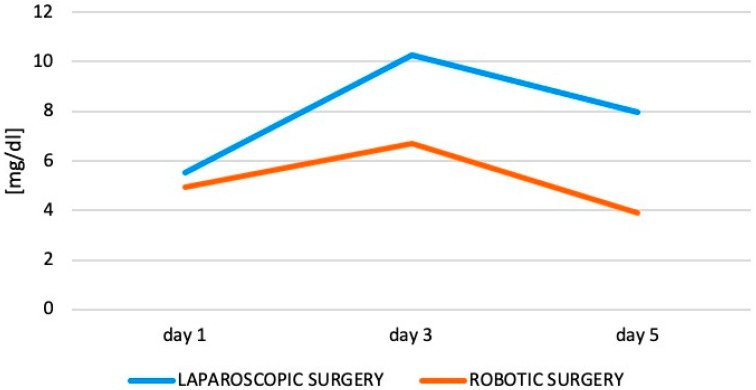
Differences in C-reactive protein (CRP) over time.

**Figure 2 ijerph-19-10606-f002:**
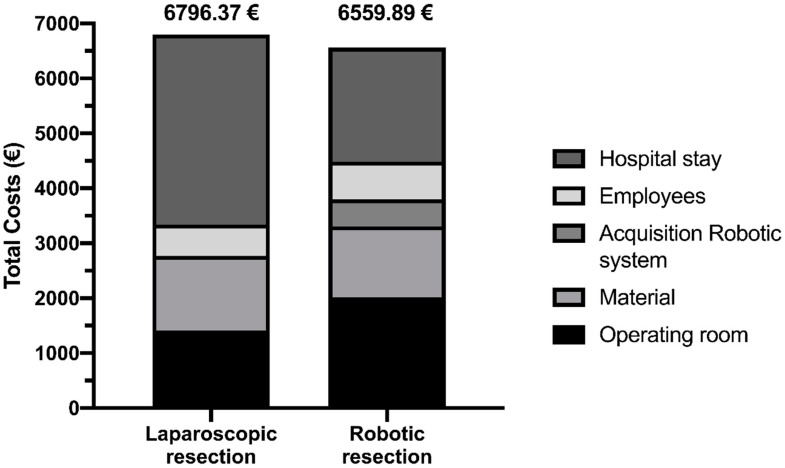
Cost-effectiveness.

**Table 1 ijerph-19-10606-t001:** Patient characteristics.

Characteristics	All	Laparoscopic	Robotic	*p*-Value
	(n = 61, 100%)	(n = 35, 57.4%)	(n = 26, 42.6%)	
Sex				0.05
Male	30 (49.2%)	21 (60%)	9 (34.6%)	
Female	31 (50.8%)	14 (40%)	17 (65.4%)	
Age at operation (years) median (range)	58 (28–88)	61 (28–88)	56 (33–83)	0.145
BMI (kg/m^2^) (range)	27.3 (17.6–48.3)	27.6 (19.5–48.3)	26.9 (17.6–36.4)	0.629
ASA classification (range)	2.2 (1–3)	2.2 (1–3)	2.1 (1–3)	0.139
CCI (range)	3.1 (0–10)	3.4 (0–10)	2.6 (0–9)	0.173
Disease				
Diverticulitis *	42 (68.9%)	23 (65.7%)	19 (73.1%)	
Malignancy ^#^	18 (29.5%)	12 (34.3%)	6 (23.1%)	
Endometriosis	1 (1.6%)	0	1 (3.8%)	
Cardiovascular disease				0.268
Coronary heart disease	2 (3.3%)	2 (3.3%)	0	
STEMI/NSTEMI	4 (6.6%)	1 (2.9%)	3 (11.5%)	
Heart failure	1 (1.6%)	1 (2.9%)	0	
Liver disease				0.095
Fibrosis/NASH	2 (3.3%)	0	2 (7.7%)	
Immunosuppression	4 (6.6%)	3 (8.6%)	1 (3.8%)	0.461
CKD				0.685
>II	7 (11.5%)	4 (11.4%)	3 (11.5%)	
Diabetes mellitus				0.431
IDDM	2 (3.3%)	2 (5.7%)	0	
NIDDM	3 (4.9%)	2 (5.7%)	1 (3.8%)	
Anticoagulation				
NGOA	1 (1.6%)	0	1 (3.8%)	0.242
ASS	10 (16.4%)	6 (17.1%)	4 (15.4%)	0.501
Dual	1 (1.6%)	0	1 (3.8%)	0.501
Operating time, min (range)		173 (102–291)	254 (150–381)	0.001
Conversion		4 (11.4%)	3 (11.5%)	0.989
Length of hospital stay, days (range)		10 (5–45)	6 (4–10)	0.025

ASA, American Society of Anesthesiologists; ASS, acetylsalicylic acid; BMI, body mass index; CCI, Charlson Comorbidity Index; CKD, chronic kidney disease; IDDM, insulin-dependent diabetes mellitus; NASH, non-alcoholic steatosis hepatitis; NGOA, new generation of oral anticoagulants; NIDDM, non-insulin-dependent diabetes mellitus; NSTEMI, non-ST-elevated myocardial infarction; STEMI, ST-elevated myocardial infarction; * chronic relapsing diverticulitis; ^#^ left-sided colon cancer UICC I–III.

**Table 2 ijerph-19-10606-t002:** Multivariate analysis for length of hospital stay (LOS).

Characteristics	*p*-Value	HR
Operation time	0.486	0.981
BMI	0.096	1.775
Surgical technique	0.095	1.782
CRP Day 3	<0.001	4.906
CRP Day 5	<0.001	4.629

BMI, body mass index; CRP, C-reactive protein.

**Table 3 ijerph-19-10606-t003:** Postoperative complications.

Characteristics	Laparoscopic (n = 35, 57.4%)	Robotic (n = 26, 42.6%)	*p*-Value
CCI, median	10.3 (0–62.5)	7.3 (0–33.5)	0.447
Clavien–Dindo > II	5 (14.3%)	1 (3.8%)	0.176
Anastomotic leakage	3 (8.6%)	0	0.126
Transfusion	2 (5.7%)	0	0.215
Re-operation	3 (8.6%)	0	0.126
MTL30	2 (5.7%)	0	0.215

CCI, comprehensive complication index; MTL30, mortality, transfer, length of stay > 30 d.

**Table 4 ijerph-19-10606-t004:** Differences of leukocytes/C-reactive protein (CRP) over time.

Characteristics	Laparoscopic (n = 35, 57.4%)	Robotic (n = 26, 42.6%)	*p*-Value
Day 1			
Leukocytes	10.86	10.26	0.51
CRP	5.54	4.94	0.56
Day 3			
Leukocytes	8.48	8.30	0.83
CRP	10.25	6.70	0.13
Day 5			
Leukocytes	7.64	8.04	0.70
CRP	7.95	3.89	0.03

CRP, C-reactive protein.

**Table 5 ijerph-19-10606-t005:** Cost-effectiveness (Euro).

	Laparoscopic	Robotic
Operating room	1412.83	2017.17
Material	1355.53	1289.93
Robotic system *		497.07
Employees	572.51	683.12
Hospital stay	3455.50	2073.20
**Overall**	**6796.37**	**6559.89**

* Costs/patient calculated for a period of 10 years with 8 operating days/month.

## Data Availability

Data are stored in an institutional database. Therefore, restrictions on availability apply due to data protection regulations. Anonymized data are, however, available from the corresponding author on reasonable request.

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
