# Peer review of "Robotic-Assisted versus Laparoscopic Left Hemicolectomy—Postoperative Inflammation Status, Short-Term Outcome and Cost Effectiveness"

_ijerph, 2022, doi:10.3390/ijerph191710606_

Round 1
Reviewer 1 Report
I would like to compliment to the authors for this well written paper.
I have just few questions to make.
1. You do not specify how many surgeons performed the operations and is not clear wheter the ones who did the laparoscopic operations were the same doing the robotics
2.If the surgeons were at the beginning of their robotic activity. In this case, you should highlight this aspect because the results would be even more significant .
3.Be more specific on the indication for the surgery (stage of the cancer, type of diverticulitis)
Reviewer 2 Report
The authors should be congratulated for this interesting study analyzing the potential role of robotic left hemicolectomy by evaluating the short-term postoperative outcomes, inflammatory status, and cost. The manuscript is clear and well written. However, a few corrections are needed before considering for publication:
- Introduction: correct the German word “und” to “and” in line 41
- Table 1: p value are not clearly understandable because of a misalignment of the last column. The authors should check if all data are correctly inserted.
- SPSS should be reported as: using IBM SPSS Statistics for Macintosh/Windows (chose which one), version 28.0 (IBM Corp., Armonk, NY, USA).
- Results: both laparoscopic and robotic group have the same conversion rate for left hemicolectomy. This result should be better discussed in the discussion section. Do the authors believe the high robotic conversion rate is a consequence of the learning curve which is not yet reached? How come a robotic approach which for example for advanced pelvic procedures as intersphincteric resection usually have 0% conversion rate here is similar? The authors need to better discuss this according to their opinion
- What is the status of the robotic learning curve of the team at the time of the study? How many surgeons performed the laparoscopic and how many the robotic? Are they the same surgeons?This should be added in the material and methods section
- It is not clear the starting of the robotic program in the center (summer 2021). Since then only the robotic approach was used? How come then there is a difference in sex between the 2 groups? Did the authors prefer the robotic approach for female patients who are easier as first robotic patients? This should be discussed because the authors reported no indication difference between the 2 approaches.
- Moreover, it would be useful to provide a figure showing the number or rate for each approach reported for each month of the year (bar chart) in order to better show the implementation of the robotic program in the department and better understand the indication and results of this study.
- The difference in anastomotic leak between lap and robotic group should be most probably consequent to the small series dimension. This was commented in the discussion section but should be reported in a stronger way otherwise there could be a misunderstanding from the audience.
- The authors reported a difference in CRP levels between the two groups. Could this be consequent to the slight difference in the treated disease (benign/malign)? Did the authors run the disease type in the multivariate analysis as risk factor for CRP levels? Is there any difference of CRP levels in a sub-analysis on only benign and only malign subgroups?
- Line 232: the word days is missing in “eight operating…. per month”
- Line 239-241: this sentence should be rephrased as it has no sense the way it is written.
